# Antibiotic Susceptibility of Bacterial Pathogens That Infect Olive Flounder (*Paralichthys olivaceus*) Cultivated in Korea

**DOI:** 10.3390/ijerph19138110

**Published:** 2022-07-01

**Authors:** Ye Ji Kim, Lyu Jin Jun, Da Won Lee, Young Juhn Lee, Ye Jin Ko, Yeong Eun Oh, Soo Ji Woo, Myoung Sug Kim, Seung Min Kim, Joon Bum Jeong

**Affiliations:** 1Department of Marine Life Sciences, Jeju National University, Jeju 63243, Korea; kyjk4150@jejunu.ac.kr (Y.J.K.); jindan97@jejunu.ac.kr (L.J.J.); liv817@hanmail.net (D.W.L.); lyz1110@naver.com (Y.J.L.); yejinny9923@naver.com (Y.J.K.); dhduddms1912@naver.com (Y.E.O.); 2Pathology Research Division, National Institute of Fisheries Science, Busan 46083, Korea; wsj2215@korea.kr (S.J.W.); fishdoc@korea.kr (M.S.K.); 3Mokpo Regional Office, National Fishery Products Quality Management, Mokpo 58746, Korea; everydayok@korea.kr

**Keywords:** antibiotic resistance, MAR index, MIC panel, aquaculture, *Paralichthys olivaceus*

## Abstract

*Paralichthys olivaceus* (olive flounder) is widely cultivated in Korea. However, data on the antibiotic susceptibility of bacterial pathogens that infect olive flounders in Korea are limited. The susceptibility of 84 strains of 3 pathogenic bacteria (*Streptococcus* spp., *Vibrio* spp., and *Edwardsiella piscicida*) to 18 antibiotics was tested using the minimum inhibitory concentration (MIC) panels, and the distribution of the MIC values for each species was confirmed. Among the panel antibiotics, nine commonly used antibiotics were selected, and the multiple antibiotic resistance (MAR) index and antibiotic resistance pattern were indicated using the disk diffusion method. It was confirmed that most of the isolates had a MAR index greater than 0.2, indicating a high-risk source. The distribution patterns of the MIC values and resistance pattern between gram-positive and gram-negative bacteria showed slightly different results. Ampicillin, erythromycin, and clindamycin were more effective against gram-positive bacteria than gram-negative bacteria. However, the MIC values of flumequine for gram-positive bacteria were higher than those of gram-negative bacteria. Through the distribution patterns of the MIC values and resistance patterns presented in this study, the need for monitoring the multidrug-resistant bacteria in aquaculture is emphasised.

## 1. Introduction

The rapid growth of the aquaculture industry with the development of aquaculture technology has increased the consumption of fish per capita. Global fish production is estimated to increase from approximately 179 million tons in 2018 to 204 million tons in 2030 [1]. *Paralichthys olivaceus* (olive flounder) is an important commercial fish species in the aquaculture industry in Korea. After establishing an artificial seed production technology in Korea, the production of olive flounder increased from approximately 1037 tons in 1990 to 41,777 tons in 2021 [2]. In particular, nearly half of the production in Korea occurs on Jeju Island, which is the main production area for olive flounder [3].

Antibiotics are used in various fields such as human medicine, veterinary medicine, agriculture, livestock, and fisheries. The decline in the marketability and the increase in the mortality of fish caused by bacterial diseases result in economic losses in aquaculture. Therefore, to minimise damage and increase fish production, many fish farms use antibiotics to control bacterial diseases. The global use of antibiotics in aquaculture in 2017 was estimated at 10,259 tons, and this amount is presumed to remain steady throughout 2017–2030 [4]. Representative bacteria that cause diseases in cultivated olive flounder are *Streptococcus* spp. among gram-positive bacteria and *Vibrio* spp. and *Edwardsiella piscicida* (formerly included in *Edwardsiella tarda*) among gram-negative bacteria [5]. In aquaculture, antibiotics are used for the treatment or prophylaxis of diseases caused by these bacteria and promote fish growth when mixed with feed; however, antibiotics are used relatively freely in aquaculture owing to the lack of strong regulations and precise guidelines. Many studies have reported that the unrestrained use of antibiotics in aquaculture has led to the pollution of the marine environment. Since most fish farms are operated using a flow-through system, the aquaculture effluents are discharged to the nearby marine environment [6,7,8,9,10,11].

Based on the concept of one health that suggests that humans, animals, and the environment are all connected, international organisations such as the Food and Agriculture Organisation, World Organisation for Animal Health, and World Health Organisation continue to monitor antibiotic-resistant bacteria. In many countries, monitoring systems such as the Danish Integrated Antimicrobial Resistance Monitoring and Research Program, Canadian Integrated Program for Antimicrobial Resistance Surveillance, and National Antimicrobial Resistance Monitoring System have been established for the surveillance and monitoring of antibiotic-resistant bacteria [12]. Studies on the use of antibiotics in Korea have been systematically conducted in many research institutions such as the Korea Disease Control and Prevention Agency, Animal and Plant Quarantine Agency, Ministry of Science and ICT, Ministry of Oceans and Fisheries, Ministry of Agriculture, Food, and Rural Affairs, Rural Development Administration, and Ministry of Food and Drug Safety. However, research on the monitoring of antibiotic-resistant bacteria or criteria for determining antibiotic resistance are still insufficient in the fishery field. During bacterial disease outbreaks in aquaculture, antibiotic susceptibility testing is important to determine the susceptibility of pathogens to antibiotics or to detect antibiotic resistance in specific bacterial isolates. Antibiotic susceptibility testing can be performed using diffusion and dilution methods. The disk diffusion test, which is the most common diffusion method, has the advantage of being inexpensive and simple compared to other methods, and it can test for a large number of bacteria and antibiotics [13,14]. The MIC panel is coated with various concentrations of antibiotics on a 96-well plate, enabling rapid and accurate antibiotic susceptibility testing. The multiple antibiotic resistance (MAR) index is an important analysis method to check the risk factors of antibiotic resistance; it is a simple, fast, and effective method [15]. It is necessary to establish standards for the use of antibiotics in aquaculture by analysing the rate of antibiotic resistance of the pathogens that infect fish, and continuous investigation and surveillance of antibiotic resistance is required.

Therefore, we aimed to establish a basis for a national antibiotic resistance monitoring system in the present study by investigating the distribution of minimum inhibitory concentration (MIC) values and resistance patterns of representative bacteria isolated from olive flounders.

## 2. Materials and Methods

### 2.1. Bacterial Isolates

*Streptococcus* spp., *Vibrio* spp., and *E. piscicida* were isolated from infected olive flounders in Jeju, South Korea. The bacteria were grown in tryptic soy broth (TSB, Difco, Detroit, MI, USA) supplemented with 2% NaCl. To isolate the bacteria, thiosulfate citrate bile salts sucrose (TCBS) agar (Difco, Detroit, MI, USA) and Salmonella Shigella (SS) agar (Difco, Detroit, MI, USA) were used as selective media. A blood agar (KOMED, Seongnam, Korea) differential medium was used to isolate haemolytic *Streptococcus* spp. The bacteria were streaked on the plate and incubated at 28 °C for 24 h, and then the morphology (size, shape, texture, and colour) of the colonies was evaluated to distinguish the bacterial species. All strains were stored at −80 °C in TSB supplemented with 2% NaCl and 20% glycerol for the experiments.

### 2.2. DNA Extraction

The bacteria were inoculated in TSB and incubated overnight at 28 °C. The DNA was extracted using the Higene^TM^ Genomic DNA Prep Kit (BIOFACT, Daejeon, Korea) according to the manufacturer’s protocol. The cultured cells (1.5 mL) were transferred to a 1.5 mL microtube, the suspension was centrifuged at 12,000× *g* for 5 min, and the supernatant was removed. The cell pellet was suspended in 300 μL of a cell resuspension solution. Lysozyme (2 μL; 100 mg/mL) was added, mixed via gentle pipetting, and incubated at 37 °C for 60 min. The suspension was centrifuged for 1 min at 14,000× *g*, and the supernatant was discarded. The pellet was resuspended in 300 μL of a cell lysis solution, and 1.5 μL of RNase A (4 mg/mL) was added following lysis at 37 °C for 30 min. The samples were cooled to room temperature for 5 min, and 100 μL of a protein precipitation solution was added to lyse the sample. The solution was vortexed vigorously for 1 min to thoroughly mix the solution. After centrifugation for 5 min at 14,000× *g*, the supernatant was transferred to new 1.5 mL microtubes, and the DNA was precipitated by adding 500 μL of 100% isopropanol. The tubes were then inverted 50 times to mix the contents. The mixture was centrifuged at 14,000× *g* for 1 min, and the supernatant was removed. The pellet was washed twice with 500 μL of 80% ethanol, and the supernatant was removed after centrifugation at 14,000× *g* for 1 min. After air-drying at room temperature for 15 min, 100 μL of a DNA hydration solution was added to the dried DNA pellet. Subsequently, the solution was vortexed for 5 s and stored at −20 °C before use in the experiment.

### 2.3. Polymerase Chain Reaction (PCR)

PCR analysis was performed after the DNA was extracted. The PCR reaction mixture comprised 1 μM of each primer, 2.5 mM of each deoxynucleoside triphosphate, 10× G-Taq Buffer, 2.5 U G-Taq DNA polymerase, and 1 μL of the respective template DNA in a 0.2 mL microtube. Distilled water was added to the reaction mixture to obtain a total volume of 20 μL. The thermocycling conditions were as follows: initial denaturation at 94 °C for 5 min, followed by cycles of 94 °C for 30 s, annealing temperature for 30 s, 72 °C for 30 s, and a final extension at 72 °C for 10 min. The primer sets used for PCR, the annealing temperature, and the number of cycles are shown in Table 1. The amplified product was separated via electrophoresis using 1× TAE buffer (40 mM Tris-acetate and 1 mM EDTA) after the addition of SYBR Safe DNA Gel Stain (10,000×; Invitrogen, Waltham, MA, USA) to the 1% agarose gel. After electrophoresis, the product size was observed using an ultraviolet light detector.

### 2.4. Antibiotic Susceptibility Testing

#### 2.4.1. Broth Dilution Test

A broth dilution test was performed using the Sensititre^®^ system (TREK Diagnostic Systems, Cleveland, OH, USA). Two Sensititre^®^ panels (KRAQ1 and CAMPY2) were used according to the manufacturer’s instructions. Each plate was in a lyophilised form of a serial dilution of antibiotics used in aquaculture including oxytetracycline, ceftiofur, flumequine, enrofloxacin, neomycin, ampicillin, amoxicillin/clavulanic acid, doxycycline, trimethoprim/sulfamethoxazole, sulfisoxazole, azithromycin, ciprofloxacin, clindamycin, erythromycin, florfenicol, gentamicin, nalidixic acid, and tetracycline. Several colonies obtained from the tryptic soy agar (TSA, Difco, Detroit, MI, USA) were collected and suspended in distilled water supplemented with 1% NaCl. The suspension was adjusted to a McFarland standard of 0.5 using the Sensititre^®^ Nephelometer (TREK Diagnostic system, Cleveland, OH, USA). According to the optimised culture conditions established previously, 100 μL of the suspension was transferred to 11 mL of cation-adjusted Mueller–Hinton broth containing TES buffer (CAMHBT, TREK Diagnostic system, Cleveland, OH, USA). CAMHBT containing 5% lysed horse blood was used for testing haemolytic *Streptococcus* spp. and *Vibrio* spp., and the colonies were suspended in CAMHBT supplemented with 1% NaCl [21]. CAMHBT alone was used for testing *E. piscicida*. The prepared bacterial solution was inoculated into each well of the MIC panels (100 μL) and sealed using a dedicated film. Subsequently, the panels were incubated at 28 °C for 24 h, and the MIC values were observed. The MIC was defined as the lowest concentration of the antibiotic that inhibited the visible growth of the bacteria. All experiments were performed in triplicate.

#### 2.4.2. Disk Diffusion Test

The disk diffusion test was performed according to the Clinical and Laboratory Standards Institute (CLSI) guidelines by selecting nine antibiotics from among the antibiotics included in the MIC panel [22]. The antibiotics disks used in this study were the following (purchased from Liofilchem): ampicillin (10 μg, AMP), gentamycin (10 μg, GEN), clindamycin (10 μg, CLI), erythromycin (15 μg, ERY), doxycycline (30 μg, DXT), tetracycline (30 μg, TET), nalidixic acid (30 μg, NAL), enrofloxacin (5 μg, ENRO), and florfenicol (30 μg, FFN). In order to indicate the antibiotic resistance pattern and MAR index, the results of the disk diffusion method were analysed using the WHONET 5.6 software (WHO, Geneva, Switzerland). For the antibiotics used in the disk diffusion method, the MAR index of the isolate was calculated following the Krumperman (1983) method (MAR index = a/b where a = the number of resistant antibiotics and b = the total number of tested antibiotics) [23]. A MAR index value greater than 0.2 indicated that the isolates are derived from a high-risk source where antibiotics were frequently used.

## 3. Results

### 3.1. Identification of Strains

A total of 84 strains were isolated from olive flounders in Jeju. The isolates were grown on TSA supplemented with 1% NaCl and TSB supplemented with 2% NaCl, and they were classified using TCBS, SS, and blood agar as the selective media. *Vibrio* spp. were confirmed as green or yellow colonies on TCBS. *E. piscicida* formed black colonies on SS. PCR was performed using specific primer sets for identifying the strains. Out of the 29 *Streptococcus* isolates, 1 isolate was identified as *S. iniae,* whereas the rest were identified as *S. parauberis* (*n* = 28). A total of 28 *Vibrio* spp. isolates were identified as *V. harveyi* (*n* = 19), *Photobacterium damselae* (*n* = 3), *V. alginolyticus* (*n* = 3), *V. parahaemolyticus* (*n* = 1), *V. anguillarum* (*n* = 1), and *V. scophthalmi* (*n* = 1). The remaining 27 strains were confirmed as *E. pisicicda* by referring to the method described by Griffin et al. (Table 2) [17].

### 3.2. Distribution of MIC Values

To determine the distribution of MIC values for bacteria that infect fish, we performed an antibiotic susceptibility test using 18 antibiotics against 84 major pathogenic bacteria that infect fish. In the graph of the MIC values derived from the experiment performed in triplicate, the second and third results are shown in dark gray and light gray, respectively. The distribution of MIC values obtained for 29 strains of *Streptococcus* spp. using the MIC panel is shown in Figure 1. Relatively low MIC values for ampicillin, amoxicillin/clavulanic acid, trimethoprim/sulfamethoxazole, clindamycin, and erythromycin were observed among the *Streptococcus* spp. In particular, the MIC values of ampicillin and amoxicillin/clavulanic acid could not be determined because most strains had MIC values lower than the test range. Most of the *Streptococcus* spp. showed low MIC values for the three tetracycline-derived antibiotics (oxytetracycline, doxycycline, and tetracycline); however, nine strains showed a distribution of high MIC values and were identified as resistant strains. In contrast, all 29 strains showed MIC values higher than the highest concentration of antibiotics tested for sulfisoxazole and nalidixic acid. Most of the *Streptococcus* spp. showed MIC values within the corresponding range against antibiotics including flumequine (32–128 μg/mL), enrofloxacin (0.12–1 μg/mL), neomycin (8–32 μg/mL), and gentamycin (2–8 μg/mL).

The *Vibrio* spp. isolates showed high MIC values for ampicillin, clindamycin, and erythromycin (Figure 2). The MIC for ampicillin is mostly 256 μg/mL or more, that of clindamycin ranged from 4 to >16 μg/mL, and that of erythromycin ranged from 2 to >64 μg/mL. Among the tetracyclines antibiotics (including oxytetracycline, doxycycline, and tetracycline), doxycycline was associated with a relatively low MIC value. The MIC values for the *Vibrio* isolates varied for all three fluoroquinolone antibiotics (flumequine, enrofloxacin, and ciprofloxacin).

*E. piscicida* showed a higher level of resistance than that of the other strains, and all 27 isolates showed MIC values higher than the test range of the MIC panel for the clindamycin of the lincosamide group (Figure 3). In addition, the *E. piscicida* isolates showed high MIC values for macrolides (azithromycin and erythromycin). Of the 27 isolates tested, 15 were confirmed to be resistant to tetracycline antibiotics, and the other 12 showed a relatively low MIC value distribution.

### 3.3. MAR Index and Antibiotic Resistance Patterns

In this study, the results of the disk diffusion method were used to indicate the MAR index and antibiotic resistance patterns. The measured MAR index range of the *Streptococcus* spp. isolate was from 0.11 to 0.67, and those of *Vibrio* spp. and *E. piscicida* ranged from 0.33 to 0.89. About 75.86% of *Streptococcus* spp. were found to be resistant to one or more antibiotics (Figure 4). *E. piscicida* was observed in 48.14% of the strains resistant to seven or more antibiotic classes. The highest MAR index of 0.89 (resistance to eight antibiotics) was observed in both *Vibrio* spp. and *E. piscicida*. As a result of analysing the resistance pattern of the isolate with the WHONET program, the *Streptococcus* spp. strain had the most NAL and GEN/NAL patterns at 24.14%, respectively. The most frequent patterns of *Vibrio* spp. and *E. piscicida* were the AMP/CLI/ERY pattern (28.57%) and the AMP/CLI/ERY/DXT/TET/NAL/ENRO pattern (44.44%), respectively (Table 3).

## 4. Discussion

Olive flounders are widely cultivated in Jeju, Korea; however, data on the antibiotic susceptibility of bacterial pathogens that infect olive flounders in Korea are limited. Bacteria that represent a threat to the aquaculture industry annually include *Streptococcus* spp., *Vibrio* spp., and *E. piscicida*. *S. parauberis* and *S. iniae* are representative gram-positive bacteria that cause diseases in cultivated olive flounder. Streptococcosis is a disease characterised by symptoms such as the darkening of body colour and abdominal distension, and it is associated with economic losses in the aquaculture industry [24]. Among the *Vibrio* genus known to contain approximately 72 species [25], the *Vibrio* spp. isolated in this study were *V. harveyi* (19 strains), *P. damselae* (three strains), *V. alginolyticus* (three strains), *V. parahaemolyticus* (one strain), *V. anguillarum* (one strain), and *V. scophthalmi* (one strain). Additionally, the gram-negative bacterium *E. piscicida,* formerly known as *E. tarda,* was isolated from olive flounders. The major clinical signs of *E. piscicida* infection in fish include rectal hernia, petechial haemorrhage in the fin and skin, and the opacity of the eyes [26,27].

Based on the MICs obtained using two panels including 18 antibiotics tested against strains isolated from olive flounders, *Streptococcus* spp. showed high levels of resistance to flumequine, sulfisoxazole, and nalidixic acid. In particular, sulfisoxazole and nalidixic acid were found to show higher MICs for *Streptococcus* spp. than the highest concentration in the range tested for all isolates used in this study. Nalidixic acid and sulfisoxazole have been reported to be ineffective against gram-positive bacteria such as *Streptococcus* spp. [28,29]. In contrast, the *Streptococcus* isolates showed low levels of resistance to ampicillin and amoxicillin/clavulanic acid.

In the distribution of MIC values, the antibiotic resistance patterns based on the isolates slightly differed between gram-positive and gram-negative bacteria. In particular, ampicillin was more effective against gram-positive bacteria than it was against gram-negative bacteria. *Streptococcus* spp. showed low MIC values for erythromycin and clindamycin, whereas *Vibrio* spp. and *E. piscicida* showed high MIC values for these antibiotics. *Streptococcus* spp. showed high MIC values for flumequine, whereas all of the gram-negative bacterial isolates (*Vibrio* spp. and *E. piscicida*) showed a high sensitivity to this antibiotic. Previous studies have shown that *S. parauberis* isolated from olive flounder showed MICs in the ranges of 0.0039 to 1 μg/mL and 0.5 to 8 μg/mL for XNL and FFN, respectively [30]. Our results showed that the MIC range of XNL was mostly 0.06–4 μg/mL, which was slightly higher than that of the previous study. The MIC range of FFN was 0.5–2 μg/mL, indicating relatively low MIC values. The ranges of MIC results against the OTC of *S. parauberis* isolated from olive flounder in Jeju by Park et al. (2018) [31] were from <0.25 to 0.5 μg/mL and from 32 to >256 μg/mL, which were similar to our results. Previous studies have reported that *Vibrio* spp. isolated from aquatic environments are resistant to ampicillin, which belongs to the β-lactam group of antibiotics [32,33,34]. In this study, 97% (27 isolates) of the 28 *Vibrio* isolates showed resistance to ampicillin. Zanetti et al. (2001) [35] suggested that AUG2 and CIP could be an effective antibiotic therapy for *V. parahaemolyticus* and *V. alginolyticus* isolated from the environment. The *Vibrio* spp. isolated in this study was mostly found in the intermediate concentration range of 4/2–32/16 μg/mL and 0.06–2 μg/mL, respectively. Doxycycline was associated with a relatively low MIC value for *Vibrio* spp., which is likely because it was recently used as a second-generation antibiotic, with minocycline introduced during the 1960s [36]. *E. piscicida* isolates showed relatively high MIC values for 15 antibiotics, excluding neomycin, ciprofloxacin, florfenicol, and gentamycin. *E. tarda* isolated from various countries (UK, USA, Germany, Sweden, Japan) were naturally susceptible to TET and DXT, with MIC values in the range from 0.5 to 2 μg/mL [37]. However, more than half of the isolates were found to have an MIC value higher than 64 μg/mL in this study, confirming that the strain isolated from Korea was more resistant than those of other countries.

As a result of analysing the antibiotic resistance pattern of representative fish disease bacteria, it was confirmed that all but two patterns had different resistance patterns. These results mean that when bacterial disease occurs in fish, the antibiotics to be selected are different for each bacterial species. Among the isolated bacteria, isolates that are resistant to two or more antibiotics were designated as the high-risk group. As a result of the 84 isolates isolated in this study, the MAR index of the remaining 77 strains, except for 7 strains (all *S. parauberis*), was identified as a high-risk group of greater than 0.2. The MAR index values of the *E. piscicida* isolated from a river in Thailand range from 0.25–0.92 [38], and the MAR index range for the *E. piscicida* isolated in this study was similar: 0.33–0.89.

In the feed residue, sediment, and fish feces present in the aquaculture environment, not only antibiotic-resistant bacteria but also antibiotic resistance genes exist. This contaminant is affected in the nearby seawater as it is discharged into the aquatic environment, untreated from the aquaculture [39]. The prevalence of antibiotic-resistant bacteria or multidrug-resistant bacteria in the aquatic environment is a major concern because it causes the problem of transmission to humans and animals [40]. In particular, there is a lack of research on antibiotic resistance and antibiotic resistance genes in the marine environment, so systematic study is needed. Cho et al. (2016) [41] confirmed the transfer characteristics of the resistance gene by the filter mating method. As a result, it was suggested that the resistance gene of Staphylococci isolated from the marine environment could be transferred to other bacteria including fish disease bacteria. However, it was not the result of a direct study of pathogenic bacteria in fish.

Continuous research on the antibiotic resistance of pathogenic bacteria that infect fish is needed because it is essential for controlling the occurrence of multidrug-resistant bacteria and for the selection of appropriate therapeutic agents. Therefore, the MIC values investigated in this study are expected to be used as basic data for antibiotic-resistant bacteria. In addition, the MAR index results suggest that bacteria (*Streptococcus* spp., *Vibrio* spp., and *E. piscicida*) isolated from fish can contribute significantly to the spread of multi-antibiotic resistance and antibiotic resistance genes in the marine environment.

In general, the guidelines of CLSI are used to determine susceptibility and resistance using MIC values. However, only one strain (*Aeromonas salmonicida*) was presented in the document published for bacteria isolated from aquatic animals by CLSI [42]. Therefore, the MIC results of the *Streptococcus* spp., *Vibrio* spp., and *E. piscicida* isolated in this study are used to establish new interpretation criteria in a bacterial disease of fish. MIC values of antibiotics for more diverse strains should be obtained in future studies. Since antibiotic resistance to pathogens that infect fish is essential for suppressing the occurrence of multidrug-resistant bacteria and selecting appropriate therapeutic agents, continuous research is required.

## 5. Conclusions

This study aimed to evaluate the antibiotic susceptibility of three pathogens that infect olive flounders cultivated in Korea. The susceptibility of 84 strains of 3 pathogenic bacteria (*Streptococcus* spp., *Vibrio* spp., and *E. piscicida*) to 18 antibiotics was tested using the minimum inhibitory concentration (MIC) panels KRAQ1 and CAMPY2, and the distribution of MIC values for each species was confirmed. The MAR index and antibiotic resistance pattern were confirmed by the disk diffusion method. Regarding the distribution of MIC values between gram-positive and gram-negative bacteria, ampicillin, erythromycin, and clindamycin were more effective against gram-positive bacteria (*Streptococcus* spp.) than gram-negative bacteria (*Vibrio* spp., and *E. piscicida*), while the MIC values of flumequine for gram-positive bacteria exceeded those for gram-negative bacteria. The antibiotic susceptibility values determined in this study are expected to serve as basic data for monitoring antibiotic-resistant bacteria that infect cultivated olive flounders in Korea.

## Figures and Tables

**Figure 1 ijerph-19-08110-f001:**
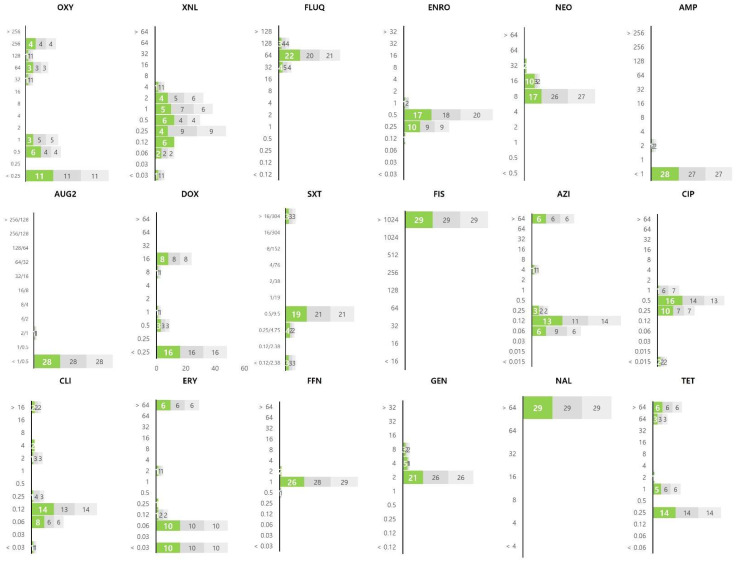
The MIC value distribution of 18 antibiotics against 29 strains of *Streptococcus* spp. isolated from olive flounders. The X and Y axes of the graph represent the number of isolates and the MIC (μg/mL), respectively. The results of the experiment repeated in triplicate are shown in colour on the graph (1st: green, 2nd: dark gray, 3rd: light gray). Oxytetracycline: OXY; ceftiofur: XNL; flumequine: FLUQ; enrofloxacin: ENRO; neomycin: NEO; ampicillin: AMP; amoxicillin/clavulanic acid: AUG2; doxycycline: DOX; trimethoprim/sulfamethoxazole: SXT; sulfisoxazole: FIS; azithromycin: AZI; ciprofloxacin: CIP; clindamycin: CLI; erythromycin: ERY; florfenicol: FFN; gentamicin: GEN; nalidixic acid: NAL; tetracycline: TET.

**Figure 2 ijerph-19-08110-f002:**
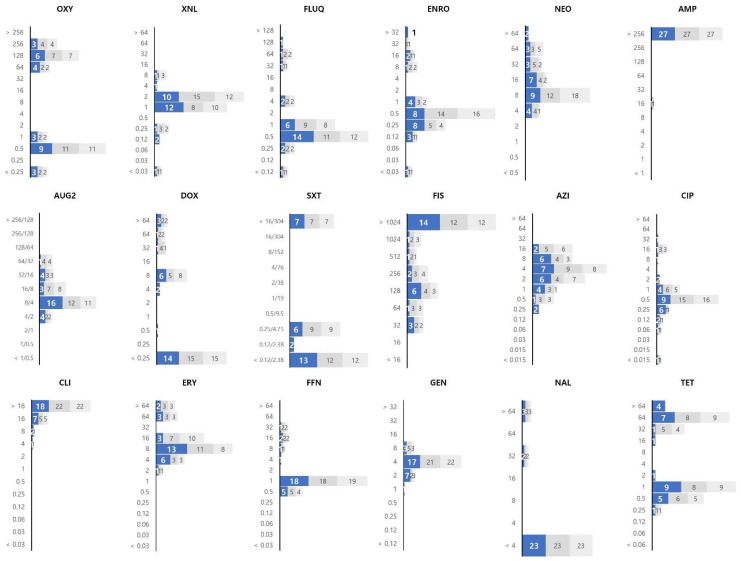
The MIC value distribution of 18 antibiotics against 28 strains of *Vibrio* spp. isolated from olive flounder. The X and Y axes of the graph represent the number of isolates and the MIC (μg/mL), respectively. The results of the experiment repeated in triplicate are shown in colour on the graph (1st: blue, 2nd: dark gray, 3rd: light gray). Oxytetracycline: OXY; ceftiofur: XNL; flumequine: FLUQ; enrofloxacin: ENRO; neomycin: NEO; ampicillin: AMP; amoxicillin/clavulanic acid: AUG2; doxycycline: DOX; trimethoprim/sulfamethoxazole: SXT; sulfisoxazole: FIS; azithromycin: AZI; ciprofloxacin: CIP; clindamycin: CLI; erythromycin: ERY; florfenicol: FFN; gentamicin: GEN; nalidixic acid: NAL; tetracycline: TET.

**Figure 3 ijerph-19-08110-f003:**
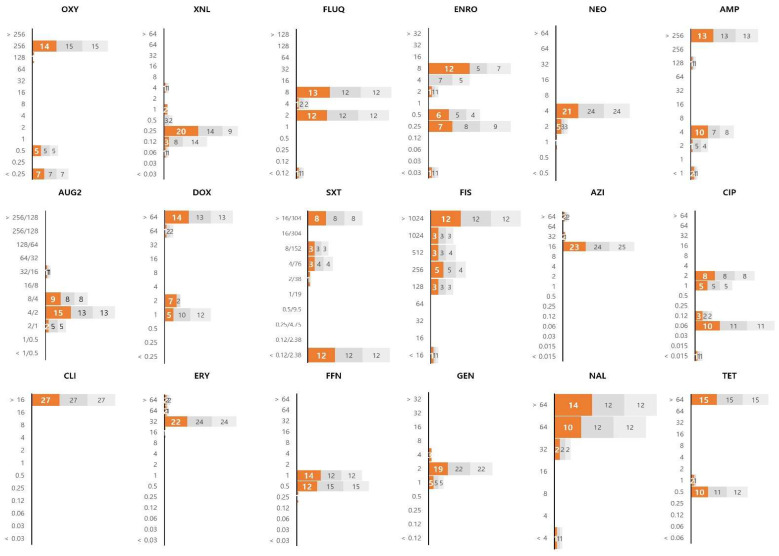
The MIC value distribution of 18 antibiotics against 27 strains of *Edwardsiella piscicida* isolated from olive flounders. The X and Y axes of the graph represent the number of isolates and the MIC (μg/mL), respectively. The results of the experiment repeated in triplicate are shown in colour on the graph (1st: orange, 2nd: dark gray, 3rd: light gray). Oxytetracycline: OXY; ceftiofur: XNL; flumequine: FLUQ; enrofloxacin: ENRO; neomycin: NEO; ampicillin: AMP; amoxicillin/clavulanic acid: AUG2; doxycycline: DOX; trimethoprim/sulfamethoxazole: SXT; sulfisoxazole: FIS; azithromycin: AZI; ciprofloxacin: CIP; clindamycin: CLI; erythromycin: ERY; florfenicol: FFN; gentamicin: GEN; nalidixic acid: NAL; tetracycline: TET.

**Figure 4 ijerph-19-08110-f004:**
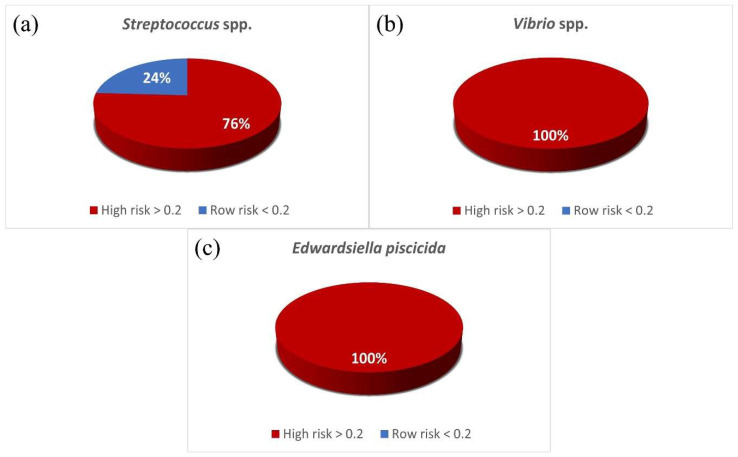
Classification of isolates by MAR index. High risk (red area) means that the isolate has a MAR index greater than 0.2 derived from a high-risk source of antibiotic contamination [23]: (**a**) *Streptococcus* spp.; (**b**) *Vibrio* spp.; (**c**) *Edwardsiella piscicida*.

**Table 1 ijerph-19-08110-t001:** Primer sets used for the detection of bacterial isolates in this study.

Target Pathogen	Oligonucleotide Sequences	Product Size (bp)	Annealing Temp. (°C)	Cycles	Reference
*Streptococcus iniae*	5′-AAGAGACGCAGTGTCAAAAG CGTTTCTTATCTTGTTACTC-3′	107	55	33	[16]
*S. parauberis*	5′-TCCAGTCTTTCGACCTTCTT CAAAGAGATGTTCGGCTTG-3′	220			
*Edwardsiella tarda*	5′-CAGTGATAAAAAGGGGTGGA CTACACAGCAACGACAACG-3′	114	58	30	[17]
*E. piscicida*	5′-CTTTGATCATGGTTGCGGAA CGGCGTTTTCTTTTCTCG-3′	130			
*Vibrio* genus	5′-GTCARATTGAAAARCARTTYGGTAAAGG ACYTTRATRCGNGTTTCRTTRCC-3′	689	60	30	[18]
*V. alginolyticus*	5′-ACGGCATTGGAAATTGCGACTG TACCCGTCTCACGAGCCCAAG-3′	199			
*V. parahaemolyticus*	5′-AGCTTATTGGCGGTTTCTGTCGG CKCAAGACCAAGAAAAGCCGTC-3′	297			
*V. anguillarum*	5′-GTTCATAGCATCAATGAGGAG GAGCAGACAATATGTTGGATG-3′	519			[19]
*V. harveyi*	5′-GTGATGAAGAAGCTTATCGCGATT CGCCTTCTTCAGTTAACGCAGGA-3′	601			[20]
*V. ichthyoenteri/V. scophthalmi*	5′-ATGCAATCATGCCTCAAGATCTA AAATGTACCTTCTTCAGTCAACTT-3′	434			
*Photobacterium damselae*	5′-CAAGACATCATCGATGTGATGCGT GAAACTTTACCATCTACCACTTTG-3′	533			

**Table 2 ijerph-19-08110-t002:** List of bacteria isolated from olive flounder in Jeju.

Species	Year	No. of Isolates
*Streptococcus parauberis*	2019	28
*S. iniae*	2015	1
*Vibrio harveyi*	2013–2017	19
*Photobacterium damselae*	2016–2020	3
*V. alginolyticus*	2018–2020	3
*V. anguillarum*	2013	1
*V. scophthalmi*	2015	1
*V. parahaemolyticus*	2020	1
*Edwardsiella piscicida*	2010	4
*E. piscicida*	2013	1
*E. piscicida*	2014	2
*E. piscicida*	2015	3
*E. piscicida*	2019	16
*E. piscicida*	2020	1
Total		84

**Table 3 ijerph-19-08110-t003:** Multiple antibiotic resistance index (MAR) and resistance pattern of bacterial isolates.

No.	Antibiotic Resistant Pattern	MAR Index (No. of Antibiotics Resistance)	% of Isolates		
			S	V	E
1	NAL	0.11 (1)	24.14	0	0
2	ERY/NAL	0.22 (2)	3.45	0	0
3	GEN/NAL		24.14	0	0
4	DXT/TET/NAL	0.33 (3)	3.45	0	0
5	ERY/TET/NAL		3.45	0	0
6	**CLI/ERY/NAL**		3.45	0	14.81
7	GEN/NAL/ENRO		6.90	0	0
8	GEN/ERY/NAL		3.45	0	0
9	AMP/CLI/ERY		0	28.57	0
10	ERY/DXT/TET/NAL	0.44 (4)	3.45	0	0
11	CLI/ERY/TET/NAL		10.34	0	0
12	CLI/ERY/DXT/NAL		3.45	0	0
13	GEN/ERY/NAL/ENRO		3.45	0	0
14	CLI/ERY/DXT/TET		0	3.57	0
15	AMP/CLI/ERY/ENRO		0	7.14	0
16	AMP/CLI/ERY/TET		0	17.86	0
17	CLI/ERY/TET/NAL		0	0	3.70
18	AMP/CLI/ERY/ENR		0	0	3.70
19	AMP/ERY/TET/NAL/ENR	0.56 (5)	3.45	0	0
20	AMP/CLI/ERY/NAL/ENR		0	10.71	0
21	AMP/CLI/ERY/TET/NAL		0	3.57	0
22	AMP/CLI/ERY/DXT/TET		0	10.71	0
23	AMP/GEN/CLI/ERY/ENRO		0	3.57	0
24	CLI/ERY/DXT/TET/NAL		0	0	3.70
25	GEN/ERY/TET/NAL/ENRO		0	0	3.70
26	AMP/CLI/ERY/NAL/ENRO		0	0	11.11
27	GEN/CLI/ERY/DXT/TET/NAL	0.67 (6)	3.45	0	0
28	AMP/CLI/ERY/DXT/TET/ENRO		0	7.14	0
29	CLI/ERY/DXT/TET/NAL/ENRO		0	0	3.70
30	AMP/CLI/ERY/TET/NAL/ENRO		0	0	3.70
31	AMP/GEN/CLI/ERY/NAL/ENRO		0	0	3.70
32	AMP/GEN/CLI/ERY/DXT/TET/ENRO	0.78 (7)	0	3.57	0
33	AMP/CLI/ERY/DXT/TET/NAL/ENRO		0	0	44.44
34	**AMP/GEN/CLI/ERY/DXT/TET/NAL/ENRO**	0.89 (8)	0	3.57	3.70

Bold text indicates overlapping resistance patterns. *Streptococcus* spp.: S; *Vibrio* spp.: V; *Edwardsiella piscicida*: E; ampicillin: AMP; gentamicin: GEN; clindamycin: CLI; erythromycin: ERY; doxycycline: DOX; tetracycline: TET; nalidixic acid: NAL; enrofloxacin: ENRO; florfenicol: FFN.

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
