# Peer review of "Antibiotic Susceptibility of Bacterial Pathogens That Infect Olive Flounder (Paralichthys olivaceus) Cultivated in Korea"

_ijerph, 2022, doi:10.3390/ijerph19138110_

Round 1

Reviewer 1 Report

In this MS, the authors investigate the antibiotic susceptibility of different bacterial pathogens infecting Paralichthys olivaceus, the result will provide some information for the aquaculture.

1.       In the method, how the morphology of the colonies was determined? Provide the reference here. The conditions for PCR ? should be described clearly. The format of ug?

2.       The experiments were performed In replicates? The quality of the figures should be further adjusted.

3.       What Is new In this MS? the new bacteria isolate was identified or the antibiotic susceptibility corresponding to the bacteria was firstly found? This should be focused.

4.       Some further experiments should be performed to investigate the effects of antibiotics on the olive flounder infected with bacteria, I think this will be best. 

5.       The English need improvement In some text. 

Reviewer 2 Report

Joon Bum Jeong and co-authors have submitted the ms ” Antibiotic susceptibility of bacterial pathogens that infect olive flounder (Paralichthys olivaceus) cultivated in Korea” for consideration for publication.

The paper is potentially interesting since data on the antibiotic susceptibility of bacterial pathogens that infect olive flounders in Korea are limited. Text is generally well written and methodologically sound; however, it has the following issues:

1. I did not find in the text how many replications of the “Broth dilution test” were performed with each isolate? If there was only replication, then doubts about the validity of the results are aroused. How were the results averaged for presentation on the graph if there were several replications? How was reflected the response variability?

2. You have got isolates obtained in different years. Could you provide any analysis of changes in bacterial resistance to antibiotics with years? Is it increased or not changed? I think it could be the most interesting part of the study.

3. Many questions for the Discussion section. It seems like a rephrasing of results. I’m expecting the explanations of why responses on some antibiotics are noticeable and on others are not? The only explanation was given for Doxycycline (lines 310-313). What about others? In addition, a comparison of your data with situations in other fish farms (from Korea or other countries) is also expected. Moreover, you can use literature data on isolates from other species for comparison.

Some minor comments that need to be addressed:

line 198 (Table 2) The fourth and fifth columns in this table are superfluous. All fish are of the same species and from the same location. This information can be given in the caption.

lines 228, 235, and 242 (Figures 2,3,4) All numbers on these figures are small and hard to read. Please, try to increase the font size for the numbers on the Y-axis. 

line 266 (Figure 5) The text about risks below the diagrams is hard to read. Please, try to increase the font size.

lines 285-292. This part of the text is more suitable for the Introduction section.

Reviewer 3 Report

The manuscript is mostly well written, well-organized and contextualized, clearly emphasizing the need for monitoring the multidrug-resistant phenomena in aquaculture, as it will be reflected in public health.

Major comments:

  1. The authors determine the MIC, yet the interpretation of the values following CLSI guidelines (S/R/I) is not as clear as it should be. As different species might have different MICs, without this information the reader could be wrongly interpreting the data. I believe the authors could include this classification in Table 2, but if they do not agree, at least clearly state it when discussing these results for all antibiotics and all pathogens. Another suggestion is to include the species-threshold for S/R (and I when applicable) for each antibiotic tested in Figures 2-4 as a dashed horizontal line, for example.
  2. Figures 2 to 4 could also include the representation of the standard deviation for the MIC determinations, as I believe the authors have done replicates, as suggested in the CLSI guidelines.
  3. I have found lines 258-260 confusing. Perhaps the authors could clarify “major” and “most pattern” in what?
  4. The wording in lines 329-333 is a bit confusing. The authors should try to rephrase it and make it clearer for the readers.

Minor comments:

  1. Throughout the manuscript, the usage of italics for species and genera should be ensured.
  2. The authors were correct by clearly choosing the antibiotics to test in concordance with the aim of the study. However, to my understanding, Figure 1 could be in the supplementary material/appendix.
  3. There is an unnecessary underline in line 329.
  4. In line 350, the phrase should not start with “And”.
  5. Sections that are not applicable should be removed (instruction for Appendix A and B). Moreover, the empty reference 37 should be deleted.

Reviewer 4 Report

Well designed collection of experiments.

Author Response

Thank you very much.

Round 2

Reviewer 2 Report

The revision significantly improved the manuscript. I think it can be published in the present form.